# Acknowledging selection at sub-organismal levels resolves controversy on pro-cooperation mechanisms

Wenying Shou*

Division of Basic Sciences, Fred Hutchinson Cancer Research Center, Seattle, United States

**Abstract** Cooperators who pay a cost to produce publically-available benefits can be exploited by cheaters who do not contribute fairly. How might cooperation persist against cheaters? Two classes of mechanisms are known to promote cooperation: 'partner choice', where a cooperator preferentially interacts with cooperative over cheating partners; and 'partner fidelity feedback', where repeated interactions between individuals ensure that cheaters suffer as their cooperative partners languish (see, for example, *Momeni et al., 2013*). However when both mechanisms can act, differentiating them has generated controversy. Here, I resolve this controversy by noting that selection can operate on organismal and sub-organismal 'entities' such that partner fidelity feedback at sub-organismal level can appear as partner choice at organismal level. I also show that cooperation between multicellular eukaryotes and mitochondria is promoted by partner fidelity feedback and partner choice between sub-organismal entities, in addition to being promoted by partner fidelity feedback between hosts and symbionts, as was previously known.

## Introduction

### Cooperation: why is it important and how can it persist?

*For correspondence: wenying. shou@gmail.com

Cooperation is wide-spread (*Axelrod and Hamilton, 1981*; *Sachs et al., 2004*; *West et al., 2006*; *Frederickson, 2013*). Cooperation is thought to drive major evolutionary transitions such as the emergence of eukaryotes and multicellularity (*Maynard Smith and Szathmary, 1998*). Cooperation between pathogenic microbes can worsen microbial infection (*Sandoz et al., 2007*; *Falsetta et al., 2014*). Cooperation between cells in tumor microenvironment can hasten cancer progression (*Rattigan et al., 2012*; *Cleary et al., 2014*). In industrial fermentation, we strive to keep microbes in a cooperative state (producing products useful to us despite the metabolic burden to microbes), and exploit inter-species cooperation to increase product yield (*Zhou et al., 2011*). Thus, understanding cooperation has broad implications in basic and applied biology.

Cooperation poses an evolutionary puzzle. A cooperator pays a cost to help other individuals. If other individuals cooperate by reciprocating, then the original cooperator may enjoy a net gain. However, other individuals are better off not reciprocating ('cheating'), in which case, the original cooperator will suffer a net loss. How might cooperation evolve and persist despite the incentive to cheat?

To state more formally, let us consider a community of interacting (Appendix note 1) individuals. A focal cooperator pays a net fitness cost (Appendix note 2) to generate benefits that can be shared or exchanged (Appendix note 3) with partners. Partners may belong to the same species as the focal cooperator (e.g. kin cooperation, *Figure 1A*) or a different species (mutualistic cooperation, *Figure 1B*). Partners may cooperate by reciprocating to the focal cooperator a similar benefit (homotypic cooperation, *Figure 1A*) or a different costly benefit (heterotypic

cooperation, *Figure 1B*) (Appendix note 4), or cheat by not reciprocating (*Figure 1A* and *Figure 1B*, unfilled blue). When cooperative and cheating partners share equal access to the focal cooperator, under certain circumstances cheating partners will always be more fit than cooperative partners (Appendix note 5). In this case, cooperation may persist only if the community is partitioned into interaction groups (groups) (Appendix note 6) that are sufficiently variable in cooperator frequency (*Price, 1970*; *Trivers, 1971*; *Bull and Rice, 1991*; *Frank, 1994*; *1997*; *Foster and Wenseleers, 2006*; *Fletcher and Doebeli, 2009*) (*Figure 2*). This way, since cooperator-dominated groups will produce more cooperative benefits than cheater-dominated groups, individuals in cooperator-dominated groups (likely cooperators) will reproduce more than individuals in cheater-dominated groups (likely cheaters). Consequently, cooperator frequency can increase in a community despite decreasing in each group (*Figure 2*).

## Current definitions of partner choice and partner fidelity feedback

Identifying orthogonal mechanisms that independently promote cooperation will help us understand contributions from different sources. This will in turn move us closer toward predicting the persistence of cooperation in a natural or an industrial system.

Several conceptual frameworks have attempted to classify pro-cooperation mechanisms (*Sachs et al., 2004*; *Nowak, 2006*; *Lehmann and Keller, 2006*; *West et al., 2007*). Unlike most frameworks which focus on kin cooperation, the framework by Sachs et al. considers both kin and mutualistic cooperation. It classifies pro-cooperation mechanisms known under a plethora of names ('kin selection', 'host sanctioning/policing', 'green beard', 'reciprocity' - to name a few) to either 'partner choice (PC)' or 'partner fidelity feedback (PFF)' (*Sachs et al., 2004*) (Appendix note 7).

In PC, a focal cooperator 'recognizes' and 'chooses' cooperative instead of cheating partners to interact with. Choice can occur before (*Figure 1C,i*) or after (*Figure 1C,ii*) interacting with partners ('conditional response') (*Sachs et al., 2004*). Archetti et al. has broadened PC to include 'screening' where a focal individual displays a single fixed response, but this response favors cooperative over cheating partners (*Archetti et al., 2011a*; *2011b*) (*Figure 1C,iii*; *Figure 3D*). Thus, PC causes cooperator frequency to vary among groups as a focal cooperator favors cooperative over cheating partners during (*Figure 1C*) or after (*Figure 3D*) the formation of interaction groups. This leaves cheating partners isolated in groups devoid of cooperators.

In PFF, individuals are "associated for an extended series of exchanges that last long enough that a feedback operates" (*Sachs et al., 2004*). That is, during repeated interactions (e.g. in a spatially-structured environment), an individual who does not help its cooperative partner will eventually suffer because its cooperative partner suffers (*Figure 1D*) or even dies. In PFF, high inter-group variability in cooperator frequency, which is required for promoting cooperation, can be achieved via population bottlenecks (drastic reductions in population sizes) (*Brockhurst and Svensson, 2007*; *Chuang et al., 2009*; *Nadell et al., 2010*; *Harcombe, 2010*; *Van Dyken et al., 2013*; *Datta et al., 2013*; *van Gestel et al., 2014*; *Waite et al., 2015*) (Appendix note 8). For example in kin cooperation, interaction groups originating from single individuals will comprise the founder and its progenies, thus exhibiting a cooperator frequency of either 1 or 0 depending on whether the founder is a cooperator or a cheater. Population bottlenecks can be realized through range expansion of a few individuals to new locations (*Hallatschek et al., 2007*; *Mitri et al., 2015*, population dilutions (*Chuang et al., 2009*), population crashes caused by cheaters (*Waite et al., 2015*), and selection for rare adaptive mutants (*Morgan et al., 2012*; *Waite and Shou, 2012*; *Asfahl et al., 2015*).

Sachs et al framework is compatible with or simplifies other conceptual frameworks (Appendix note 9). For example in kin cooperation, Hamilton's "'kin discrimination' and 'viscous population' (*Hamilton, 1964a*; *Hamilton, 1964b*) would correspond to PC and PFF, respectively. To summarize current knowledge, PC promotes cooperation through 'active' recognition/screening by a focal individual. In contrast, PFF promotes cooperation through repeated interactions among group members in groups where cooperator frequencies vary significantly due to stochasticity.

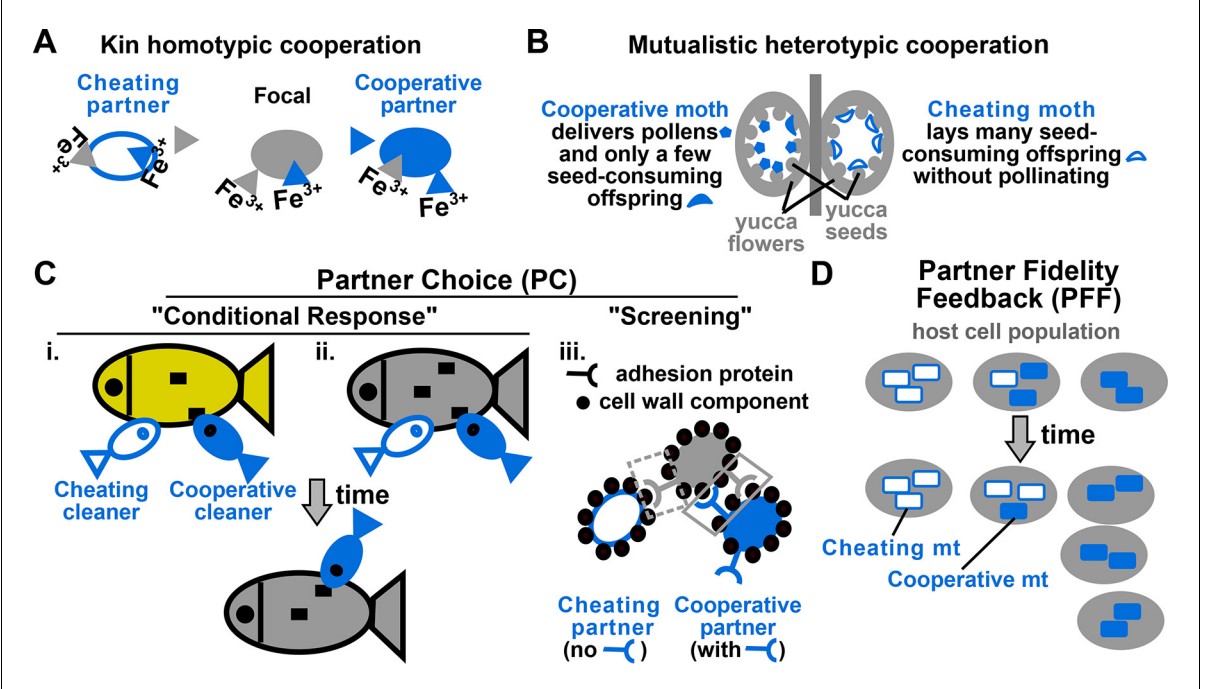

**Figure 1.** Examples of cooperation, cheating, partner choice (PC), and partner fidelity feedback (PFF). (**A**) An example of homotypic (sharing an identical benefit) and kin (between genetic relatives) cooperation, and cheating. During iron limitation, a focal cooperative *Pseudomonas aeruginosa* cell (grey filled oval) pays a fitness cost to synthesize siderophores (grey triangles, color indicating origin) which are released into the environment to scavenge iron. Siderophore-$Fe^{3+}$ complex can be taken up by the focal bacterium and partner cells. A cooperative partner (blue filled oval) also contributes siderophores which can be used by the focal cooperator. In contrast, a cheating partner (blue unfilled oval) uses siderophores without contributing any, and is competitively superior to cooperators (*Griffin et al., 2004*). (**B**) An example of heterotypic (exchanging different benefits) and mutualistic (between different species) cooperation, and cheating. In the obligatory cooperation between a yucca (grey) and yucca moths, moths can cooperate (left) or cheat (right). (**C**) In PC, a focal individual (grey) displays different responses ('conditional response', i and ii) or a fixed response ('screening', iii) that favors cooperative over cheating partners. (i and ii) A focal client (grey, bottom) will preferentially visit a cleaner (smaller blue fish) if the cleaner previously cooperated with (filled blue) instead of cheated (unfilled blue) another client (yellow, top) (*Bshary and Grutter, 2006*) or the focal client itself (grey, top) (*Bshary and Schäffer, 2002*). Cooperative cleaners only eat parasites (black squares), while cheating cleaners bite off nutritious client tissue. Thus, PC can operate before (i) or after (ii) interactions with partners. (iii) A focal cooperative yeast cell (grey) pays a fitness cost to express cell surface adhesive proteins. The focal adhesive cell will bind better to other adhesive cooperators (filled blue) than to non-adhesive cheaters (unfilled blue). This differential binding (solid versus dashed grey box) favors cooperation between adhesive cells, and allows the formation of cell clumps with enhanced stress resistance (*Smukalla et al., 2008*). (**D**) Cooperation between hosts (grey) and endosymbionts such as mitochondria (blue) has been traditionally used to illustrate PFF (*Sachs et al., 2004*; *2011*). Cooperative mitochondria (filled blue) serve the host cell at a cost to their own reproduction, and are therefore displaced by cheating mitochondria (unfilled blue) in the same cell (center panel). However, vertical transmission of mitochondria means that host and mitochondria repeatedly interact with each other. This ensures that mitochondria will harm their own fitness if they do not serve their host (compare left with right). Consequently, cooperative mitochondria can persist (compare bottom with top) if their frequency varies greatly among different hosts. For this and all following figures, filled and unfilled symbols differentiate cooperative versus cheating partners.

## Ambiguities associated with current interpretations of PFF and PC

What differentiates PC and PFF? Sachs et al. pointed out that unlike PFF, PC does not require repeated interactions between individuals (*Sachs et al., 2004*). This is certainly true. However, PC can lead to PFF (e.g. a client fish returning again and again to the same cooperative cleaner fish [(*Bshary and Schäffer, 2002*)]), thus blurring the distinction between PC and PFF.

Sachs et al. and other further suggested that unlike PC, PFF is 'automatic', 'passive', and 'does not require recognition or conditional response' (*Sachs et al., 2004*; *Foster and Wenseleers, 2006*). However, definitions of these terms are unclear. For example, a biochemist might interpret 'passive' as not requiring ATP (e.g. passive diffusion), while a geneticist might interpret 'passive' as not requiring genes (e.g. death from some environmental assaults does not require gene activities). To further illustrate this problem, let's examine PFF acting alone in the absence of PC in a community of engineered yeast strains (*Momeni et al., 2013*). In this community, adenine-requiring cooperators

release lysine ([A− L+]) while lysine-requiring partners cooperate by reciprocating adenine ([L− A+]) or cheat by not reciprocating adenine ([L−]) (*Figure 3A*). Supplying metabolites is costly (e.g. [L−] is more fit than [L− A+]) (*Waite and Shou, 2012*; *Momeni et al., 2013*). PC does not operate here: lysine released from a focal [A− L+] cell is available to both [L−] and [L− A+]. In a spatially-structured environment (e.g. on an agarose pad) without adenine or lysine supplements, [A− L+] and [L− A+] physically mix and grow to form tall 'hills' while cheating [L−] segregate to 'foothills' and fail to grow tall (*Figure 3A*). Consistent with this being PFF, disruption of repeated interactions via frequent mixing favors cheater [L−] over cooperator [L− A+]. Furthermore, population bottleneck during spatial range expansion (*Hallatschek et al., 2007*; *Mitri et al., 2015*), which presumably increases variability in cooperator frequency across different locations, favors cooperator (L− A+) over cheater (L−) (*Momeni et al., 2013*). However one could argue, albeit somewhat absurdly, that this PFF relies on the fact that cells can 'recognize' metabolites from partners via permeases and mount 'conditional responses' (growth versus no growth to cooperative versus cheating partners). And this process is not 'passive' either, because cell growth requires energy and numerous genes.

Difficulties in distinguishing PFF and PC can lead to controversy, especially when interactions occur in a spatially-structured environment where both PFF and PC can operate. In fact, mutualisms such as those between fig trees and fig wasps (*Jander and Herre, 2010*), between yuccas and yucca moths (*Pellmyr and Huth, 1994*), and between legumes and rhizobia (*Kiers et al., 2003*) have been thought to be stabilized by PC by some authors (*Sachs et al., 2004*; *Kiers et al., 2011*) and PFF by other (*Archetti et al., 2011a*; *Weyl et al., 2010*). For example, in yucca-yucca moth mutualisms, a yucca selectively aborts flowers dominated by cheating pollinator offspring (*Pellmyr and Huth, 1994*). The PC camp argues that a plant partitions its resource into separate flowers, and 'decides' whether a flower has enough cheating pollinators to be aborted (*Sachs et al., 2004*). The PFF camp counters that flower abortion is not PC, but rather the fitness consequence of being damaged by cheating pollinators during excessive oviposition, since experimentally-inflicted damages also trigger flower abortion (*Weyl et al., 2010*; *Archetti et al., 2011a*; *Weyl et al., 2011*).

This controversy has led to the proposal of an alternative criterion for distinguishing PFF and PC: If an individual's response is specific to the cheating behavior of a partner, then PC operates; if the response is toward general stress (which may or may not be inflicted by a cheating partner), then PFF operates (*Archetti et al., 2011a*). However, this alternative criterion can also encounter difficulties. Suppose that PC initially did not exist. It would seem reasonable that an evolved response to cheater-specific signals should be channeled to preexisting stress-response pathways which are already in place to facilitate survival. That is, flower abortion in response to physical damages does not preclude the possibility that the same flower abortion pathway can also be activated by cheater-specific signals. Thus, a clear criterion to differentiate PFF and PC is, to date, lacking (Appendix note 10).

## Defining PC and PFF in terms of 'entities'

In hierarchically-organized biological systems, interactions and selections can occur at multiple (including sub-organismal) levels. The controversy above arises from attempting to compress a multi-level process into a single-level process. If instead we consider 'entities' at multiple levels and allow PC and PFF to act on these entities, then this controversy is resolved.

### Definition of entity

*A biological entity (entity) is a biological structure with a boundary such that the birth, or growth, or survival, or death of an entity is separable from that of other similar entities due to chemical or physical coupling within an entity and the lack of equivalent coupling between entities* (Appendix note 11). An organism such as a yucca is an entity, because different parts of the yucca are coupled not only physically but also chemically (e.g. metabolism), and because equivalent intra-yucca coupling does not exist between yuccas. Consequently, the birth, or growth, or survival, or death of a yucca can be separated from other similar entities (Appendix note 12). Ascending above organism, an entity can be a collection of organisms. For example, a yucca and its internal moth offspring together can be considered an entity (Appendix note 13). This is because moth offspring depend on host yucca for survival, and this dependency does not extend to another yucca. The collection of moth offspring inside a yucca is also an entity, because the survival of these moth offspring are

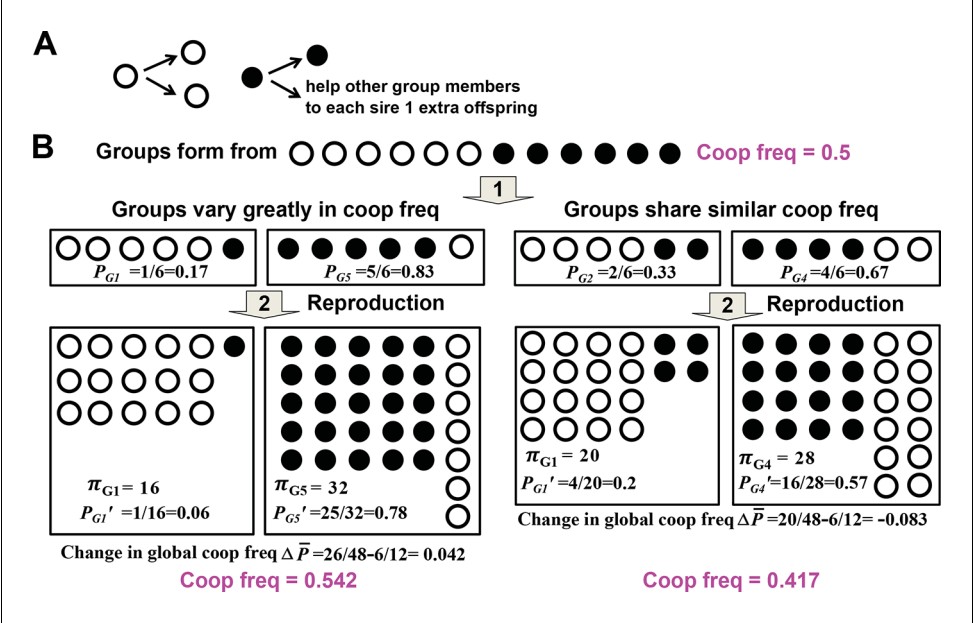

**Figure 2.** Variability in group cooperator frequency promotes cooperation. (**A**) Rules of interactions: A cooperator (filled circle) gives birth to a cooperator and helps each of the rest of group members to sire one additional offspring before dying. A cheater (unfilled circle) gives birth to two cheaters and offers no help to other group members before dying. (**B**) Community-wide cooperator frequency can increase over time when groups vary significantly in their cooperator frequencies (left), but not when they share similar cooperator frequencies (right). Suppose that after group formation and interactions and reproduction of individuals in a group, a group with initial cooperator frequency $P$ will have final size $\pi$ and final cooperator frequency of $P'$. The Price equation states that $\overline{\pi}\Delta\overline{P} = Cov(\pi, P) + Ave(\pi\Delta P)$, where $\overline{\pi}$ is the average final group size, $\Delta\overline{P} = \overline{P'} - \overline{P}$ is the difference between $\overline{P'}$ and $\overline{P}$, the final and initial community-wide cooperator frequency, respectively. $Cov(\pi, P)$, the covariance between final group size and initial cooperator frequency of group, should be positive. It may be rewritten as $\beta Var(P)$, and should increase as $Var(P)$, the variance in initial cooperator frequency $P$ across groups, increases. $Ave(\pi\Delta P)$ reflects $\Delta P$, the change in cooperator frequency in each group, and should thus be negative. Consequently, for community-wide cooperator frequency to increase, the absolute value of $Cov(\pi, P)$ must exceed that of $Ave(\pi\Delta P)$. A numerical demonstration of the Price Equation is provided below. In the case on the left, two groups form. The group starting with 1 cooperator ($G_1$) has an initial cooperator frequency of $P_{G1}$=1/6, and ends up with final group size $\pi_{G1}$= 16 and final cooperator frequency $P_{G1}'$=1/16. The group starting with 5 cooperators ($G_5$) has an initial cooperator frequency of $P_{G5}$=5/6, and ends up with final group size $\pi_{G5}$= 32 and final cooperator frequency $P_{G5}'$=25/32. The average group size $\overline{\pi}$ is (16+32)/2=24 while the change in global cooperator frequency is $\Delta\overline{P}$=26/48−6/12=0.042. Thus, $\overline{\pi}\Delta\overline{P}$ is 1.
$Cov(\pi, P) = E[(\pi - \overline{\pi})(P - \overline{P})]$, where $E[X]$ represents the expected value of $X$. Here, 1 out of 2 groups started with 1 cooperator and 1 out of 2 groups started with 5 cooperators. Thus, $Cov(\pi, P) = 0.5\left(\pi_{G1} - \overline{\pi}\right)(P_{G1} - \overline{P}) + 0.5\left(\pi_{G5} - \overline{\pi}\right)(P_{G5} - \overline{P}) = 0.5(16 - 24)\left(\frac{1}{6} - \frac{26}{48}\right) + 0.5(32 - 24)\left(\frac{5}{6} - \frac{26}{48}\right) = 2.67$; $Ave[\pi (\Delta P)] = 0.5\pi_{G1} (P_{G1}' - P_{G1}) + 0.5\pi_{G5} (P_{G5}' - P_{G5}) = 0.5 \times 16 \times \left(\frac{1}{16} - \frac{1}{6}\right) + 0.5 \times 32\left(\frac{25}{32} - \frac{5}{6}\right) = -1.67$. Hence, $Cov(\pi, P) + Ave(\pi\Delta P)$ is also 1, equal to $\overline{\pi}\Delta\overline{P}$. This figure is simplified from a lecture given by Prof. Benjamin Kerr (University of Washington, Seattle).

coupled through the survival of their shared host yucca, and such coupling does not exist with moth offspring in another yucca. Similarly, an ant colony where the task of maintaining the colony is divided among different castes is also an entity. In contrast, a collection of trees in a forest of similar trees does not constitute an entity, because coupling within a collection (e.g. competition between trees within the collection) extends to between collections. Descending below organism, an entity can also be an organismal part. For example, a yucca flower is an entity because it can be aborted independently of other flowers. Similarly, a yucca flower and its internal moth offspring together can be considered an entity, and the collection of moth offspring inside a yucca flower can also be considered an entity. Cells or gametes are entities in a multicellular organism, because they can divide and/or die independently of other similar entities. Similarly, mitochondria are entities in a cell, and mitochondrial genomes (mtDNAs) are independently replicating entities in a mitochondrion (see section below). Thus, an entity (e.g. a yucca-and its internal moth larvae) can contain smaller entities (e.g. the moth larvae; the yucca) which contain even smaller entities (e.g. flowers) which contain even smaller entities (e.g. cells, mitochondria, mtDNAs), reflecting the hierarchical organization of multicellular organisms (Appendix note 14).

## Revised definition of PC

*PC requires that a focal cooperative entity can direct more cooperative benefits to cooperative over cheating partner entities despite their spatial equivalence* (Appendix note 15) (**Figure 3B** left, **Figure 3C**). To a focal entity, partner entities are *spatially-equivalent* if the focal entity would have interacted with partner entities equally had they been identical. For example, partner entities of equal distance to a focal entity can be spatially-equivalent, so are partner entities in a well-mixed environment. As discussed earlier, PC may be achieved through conditional response of the focal entity or screening by the focal entity. I emphasize that PC must occur between a focal cooperative entity (instead of a focal entity population) and partner entities. Otherwise, PFF could be mis-interpreted as PC (Appendix note 16). In essence, my definition of PC can serve as a definition for terms such as 'active' or 'recognition'.

## Revised definition of PFF

Let us now consider potential mechanisms that promote cooperation when PC does not exist or does not operate fully (Appendix note 17) to exclude all cheaters. One mechanism orthogonal to PC is PFF. *PFF requires the formation of interaction groups that vary significantly in cooperator frequency due to stochasticity and that last long enough so that cooperators benefit from their cooperative acts* (**Figure 3B** right, **Figure 3C**). PFF can operate within a population of entities (e.g. homotypic cooperation in **Figure 3D**) or between populations of entities (e.g. heterotypic cooperation in **Figure 1D** and **Figure 3A**). As an example of PFF, let's consider 'proportional tit for tat (pTFT)', a game theory strategy. In pTFT, the focal individual starts cooperating and subsequently cooperates with a probability equal to the fraction of cooperative partners from the previous round of interactions (**Hilbe et al., 2014**). PC does not operate in pTFT, since the focal individual does not differentiate cooperative versus cheating partners in the same interaction group. Indeed, for pTFT to survive cheaters, both requirements of PFF must be satisfied: repeated interactions in groups and large variation in the frequency of pTFT across groups (**Axelrod and Hamilton, 1981**).

PC and PFF can act synergistically, an idea compatible with existing mathematical framework (e.g. Eq. 1a in [**Foster and Wenseleers, 2006**]) and can be found in the anti-competition cooperation among bacteria (**Figure 3D**) (**Chao and Levin, 1981**).

## Experimental tests of PC and PFF

The revised definition of PC leads to a conceptually simple experimental test. One can place partner entities of varying cooperative qualities at the same distance to the focal entity or together in a well-mixed environment. One can then test whether the focal entity preferentially directs benefit to the cooperative instead of cheating partner entity. If so, then PC operates.

To experimentally demonstrate the contribution of PFF in promoting cooperation, one will need to compare cooperator frequency when both PFF and PC are operative (e.g. in a spatially-structured environment) with when only PC is operative (e.g. in a well-mixed environment). Such an example is provided in **Figure 3D**. However, this experiment is sometimes not possible to do. For example, if PC also relies on a spatially-structured environment (e.g. a focal entity senses spatial gradients of benefits emanating from partner entities), then a well-mixed environment will destroy both PFF and PC. In this case, one can compare cooperator frequency when PFF acts alone (in a spatially-structured environment where PC has been mutationally inactivated) with when neither PFF nor PC is operative (in a well-mixed environment with PC mutationally inactivated).

Ideally, one would quantify processes involved in PC and PFF, and mathematically model how they contribute to cooperator survival. I illustrate how to do so using the bacterial anti-competition cooperation as an example (**Figure 3D**, Appendix note 18). If predictions on cooperator frequency match experimental observations without 'tweaking' experimentally-measured model parameters, then there is no need to invoke additional pro-cooperation mechanisms. Otherwise, we will need to look deeper into potential causes of model-experiment mismatch.

## Revisiting yucca-yucca moth cooperation

We now return to the cooperation between yuccas and yucca moths. Selection against cheating pollinators can occur *via PC between a focal yucca and its internal pollinator offspring* (**Figure 3E**, left). This is because despite the spatial equivalence of pollinator offspring (i.e. all in the yucca), the focal

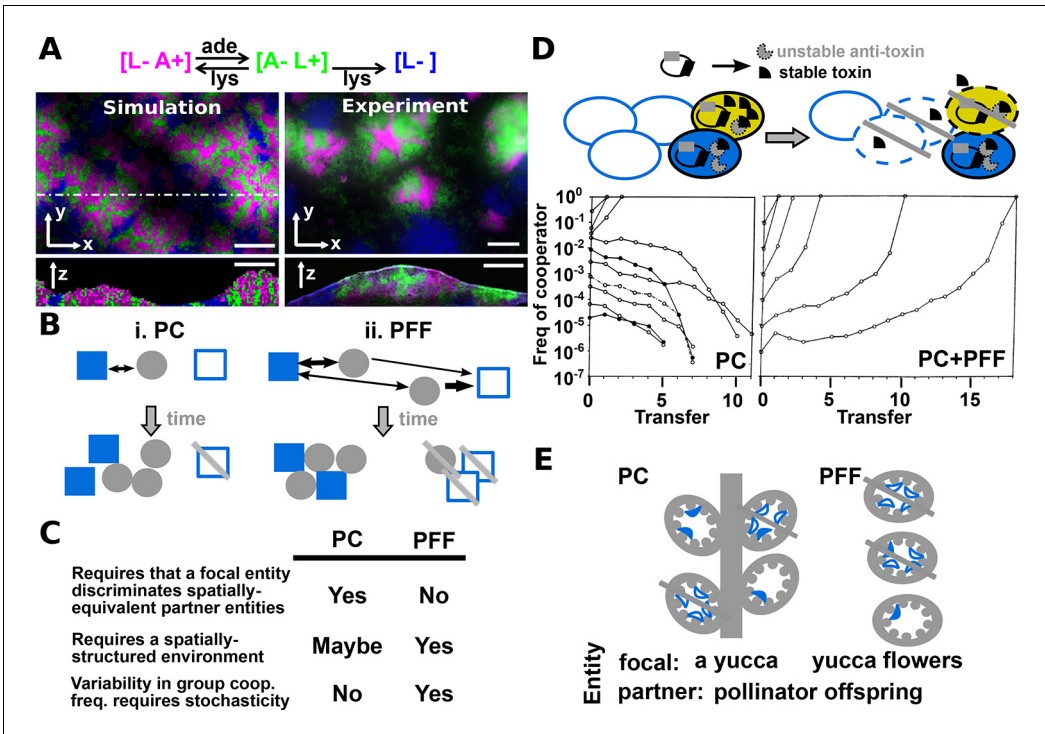

**Figure 3.** Revised definitions of PC and PFF. (**A**) Adapted from *Figure 3* in (*Momeni et al., 2013*) by B. Momeni. [L− A+], [A− L+], and [L−] are reproductively isolated yeast cells expressing different fluorescent proteins. On an agarose pad lacking adenine and lysine supplements, [A− L+] (green) and its cooperative [L− A+] (magenta) and cheating [L−] (blue) partners undergo self-organization from an initially random spatial distribution. [A− L+] and [L− A+] physically associate with each other and grow tall. [L−], isolated from [A− L+], fail to grow tall. 'xy': top-views; 'z': vertical sections. In simulated top-views, higher color intensity indicates a greater number of cells. In simulated vertical cross-sections, low and high color intensities represent dead and live cells, respectively. Scale bar: 100 μm. (**B**) In PC (i), a focal cooperative entity (grey filled circle) preferentially directs cooperative benefits to cooperative over cheating partner entities (filled and unfilled blue squares, respectively) despite their spatially equivalence. In PFF (ii), such discrimination does not exist. Instead, cooperative benefits are more available toward partner entities closer-by regardless of their cooperativeness. Beneficial interactions are marked by black arrows, with directions of benefit flow marked by arrowheads and interaction strength marked by line thickness. (**C**) A comparison of PC and PFF. (**D**) Top: Anti-competition cooperation in bacteria. Cooperators (filled ovals) but not cheaters (unfilled ovals) harbor toxin-antitoxin gene pair. Cooperators grow slower than cheaters. When encountering stress such as high cell density, a fraction of cooperators will 'commit suicide' (yellow) by lysing themselves (dashed outline) and release stable toxin (black). The remaining cooperators (filled blue) are immune to toxin because they express antitoxins (grey), while cheaters (unfilled blue) are sensitive to toxin-killing (dashed outline). This cell killing reduces competition, a benefit shared among surviving cooperators (filled blue) and cheaters (unfilled blue, solid outline). Bottom left: When PC acts alone in a well-mixed environment, cooperators can increase in frequency only if their initial abundance is sufficiently high. To see how this works, suppose that whenever total population size has reached 250, 10% of cooperators will commit suicide, and each suicidal cooperator can kill 2 cheaters. If starting at 210 cooperators and 40 cheaters, all cheaters will be killed off. However, if starting at 50 cooperators and 200 cheaters, the cheater population will diminish by 10 individuals (5%) only, compared to the 10% reduction in cooperators. Bottom right: In a spatially-structured environment, the joint action of PC and PFF favors cooperators even when cooperators are initially very rare. Data figures are reproduced from Chao and Levin's work (*Chao and Levin, 1981*) with full permission from the authors. (**E**) Yucca flowers (grey ovals) dominated by cheating (unfilled blue crescent) but not cooperative (filled blue crescent) pollinator offspring are aborted. This mechanism can be interpreted as PC between a focal yucca plant and its internal, spatially-equivalent pollinator offspring (left) or PFF between yucca flowers and pollinator offspring (right). For simplicity, pollens delivered by cooperative moths are omitted from this figure. Grey cross-bar: death of an entity.

yucca can selectively abort flowers dominated by cheating pollinator offspring. However, we can also view this as *PFF between yucca flowers and pollinator offspring* (*Figure 3E*, right). Note that since a focal flower can not discriminate its internal, spatially-equivalent pollinator offspring, PC does not operate here. Instead, PFF between flowers and pollinator offspring requires the spatially-structured environment defined by flowers (Appendix note 19). In summary, a single mechanism (aborting cheater-dominated flowers) can be viewed as PC between a focal yucca and its internal pollinator offspring *and* as PFF between yucca flowers and pollinator offspring. Even though Sachs et al. and Weyl et al. respectively classify this cooperation as being stabilized by PC and PFF, my revised definition suggests that both are correct depending on entities under consideration.

Thus, to understand and potentially predict how persistent cooperation is, we will need to consider entities at all relevant levels. Between entities at given levels, we should be mindful that PC and PFF may independently promote cooperation (*Figure 3D*). At the same time, since PFF at lower levels (e.g. between yucca flowers and moth offspring) can appear as PC at higher levels (e.g. between a focal yucca and moth offspring) (*Figure 3E*), we should avoid double-counting.

## PFF and PC in mitochondrion-eukaryote mutualistic cooperation

I will use the mutualistic cooperation between multicellular eukaryotes and mitochondria (Appendix note 20) as an example to illustrate PFF and PC at various levels. A cooperative mitochondrial genome (mtDNA) harbors genes that are necessary for the host eukaryote, but the transcription of these genes can slow down the replication of mtDNA itself because these two processes compete for the same protein factor(s) (*Larsson et al., 1998*). Thus, mtDNAs not performing host-serving functions can (*Hofhaus and Gattermann, 1999*; *Taylor et al., 2002*; *Harrison et al., 2014*; *Jasmin and Zeyl, 2014*) (Appendix note 21), though not always (Appendix note 22), gain an advantage in self-replication compared to normal mtDNAs. Here, I use 'dysfunctional mtDNAs' to describe mtDNAs that, compared to functional mtDNAs in the same 'heteroplasmic' mixture, can replicate themselves at least as well but generate less or no benefit for the host eukaryote. Dysfunctional mtDNAs can be devastating to the host eukaryote (*Chinnery et al., 2000*; *Chan, 2006*). In a heteroplasmic mixture of functional and dysfunctional mtDNAs (Fig 4A), what mechanisms oppose dysfunctional mtDNAs and their dysfunctional mitochondria? (Appendix note 23)

The importance of identifying PFF and PC at various levels becomes evident if we are interested in mitochondrial disease prognosis. Consider a human mother who is heteroplasmic for functional and dysfunctional mtDNAs/mitochondria (*Chinnery et al., 2000*). Since her children would only inherit mitochondria from her (*Pyle et al., 2015*), what would their disease prognosis be? Suppose that dysfunctional mtDNAs/mitochondria are purged from the human population solely because they harm the fitness of their host organisms (Appendix note 24) (i.e. *via PFF between host organisms and mtDNAs/mitochondria*, *Figure 4B*), as traditionally thought (*Sachs et al., 2004*; *2011*). Then, offspring should on average suffer a similar or a severer fitness defect than their mother does, since dysfunctional mtDNAs/mitochondria accumulate and are not selected against in the mother. Fortunately, as I will discuss below, PFF and PC at sub-organismal levels create somatic and germline 'filters' against dysfunctional mtDNAs/mitochondria (*Figure 4C–E*), thus increasing the fitness of the mother and her offspring.

Starting from the lowest levels, selection against dysfunctional mtDNAs can occur *via PFF between mitochondria and mtDNAs* (*Figure 4C*) (see Appendix note 25). Dysfunctional mtDNAs can be selected against if they harm the reproduction of their host mitochondria. Indeed, during *Drosophila melanogaster* oogenesis (*Hill et al., 2014*; *Ma et al., 2014*), mitochondria dominated by functional mtDNAs proliferate faster than mitochondria dominated by dysfunctional mtDNAs (*Figure 4C*). This could occur if, for example, dysfunctional mtDNAs render their host mitochondria incapable of importing proteins required for mtDNA replication.

Moving one level up, *PC can operate between a focal cell and mitochondria/mtDNAs when the cell preferentially imports, inherits, or retains functional mitochondria* (*Figure 4D*). An example of PC can be found between an oocyte and spatially-equivalent nurse cell mitochondria during *D. melanogaster* oogenesis: To be transported into the developing oocyte, mitochondria from the surrounding nurse cells need to be localized to the cytoplasmic bridge connecting nurse cells to the oocyte, which in turn depends on mitochondrial function (*Cox and Spradling, 2003*; *Hill et al., 2014*) (*Figure 4D*, left). A second example can be found during cell division when the stem daughter cell

or the young daughter cell preferentially inherits young (functional) mitochondria (*McFaline-Figueroa et al., 2011*; *Katajisto et al., 2015*), despite the spatial equivalence of mitochondria in the mother cell (*Figure 4D*, center). A third example of PC is a host cell using mitophagy, cellular digestion of mitochondria, to selectively eliminate dysfunctional mitochondria while retaining functional mitochondria (*Kim and Lemasters, 2011*; *Ashrafi and Schwarz, 2013*) (*Figure 4D*, right) (Appendix note 26). Mitophagy-mediated elimination of dysfunctional mtDNAs/mitochondria can conceivably operate in germ cells, given that mitophagy is active in fertilized oocytes to eliminate paternal mitochondria (*Ashrafi and Schwarz, 2013*). As discussed earlier (*Figure 3E*), PFF mechanisms at a lower level (i.e. between mitochondria and mtDNAs) may appear as PC mechanisms at a higher level (i.e. between a cell and mitochondria/mtDNAs). For example, slower net proliferation of dysfunctional mitochondria (*Figure 4C*) may or may not be linked to mitophagy (*Figure 4D*). An important future challenge is to address the orthogonality of various mechanisms.

For those dysfunctional mtDNAs/mitochondria that have escaped a cell's PC capability, they can be further selected against *via PFF between cells and mtDNAs/mitochondria* (*Figure 4E*). For example, in at least several types of mammalian cells, certain mutations in mtDNA will lead host cells to undergo programmed cell death (apoptosis) (*Zamzami et al., 1995*; *Chomyn and Attardi, 2003*) or cell cycle arrest (*Arnould et al., 2002*; *Owusu-Ansah et al., 2008*). At least some of these PFF

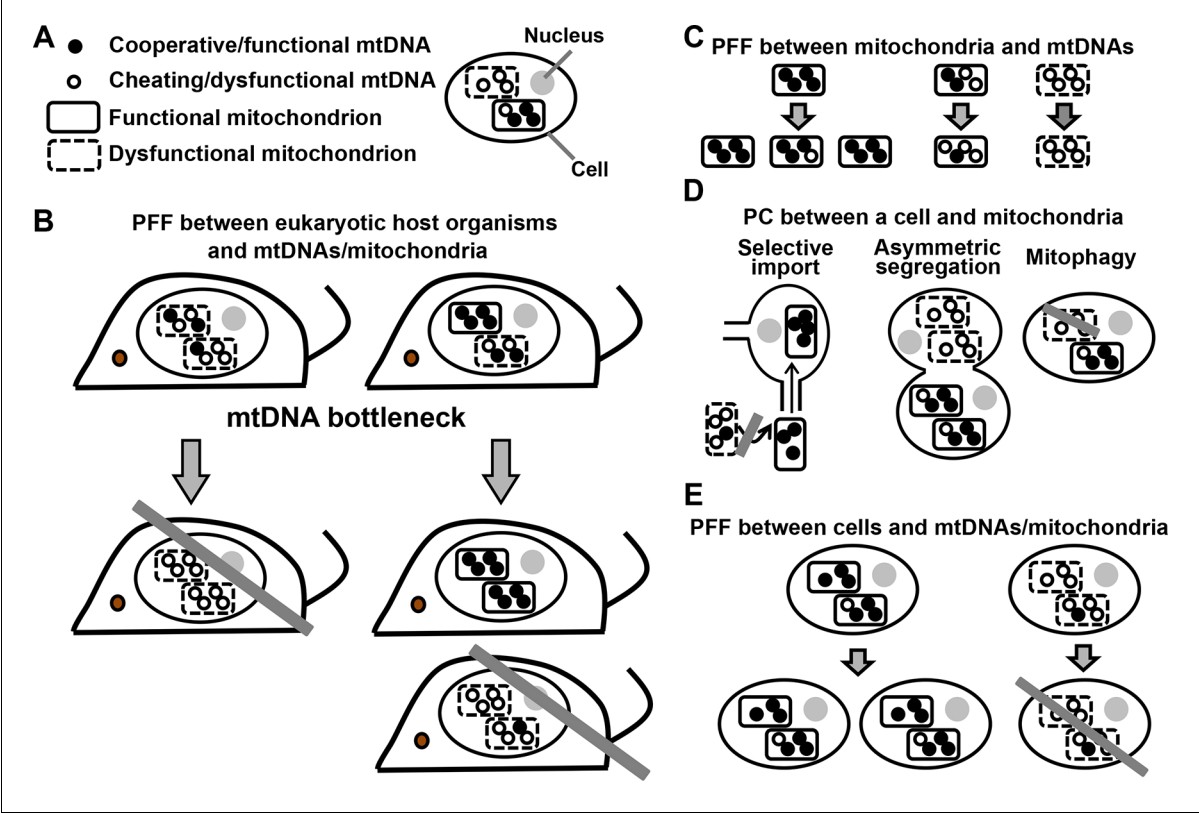

**Figure 4.** PFF and PC act at multiple levels in the mutualistic cooperation between mitochondria and multicellular eukaryotes. (**A**) An eukaryotic cell contains multiple mitochondria (dashed versus solid boundaries representing dysfunctional versus functional mitochondria). Each mitochondrion contains multiple mtDNAs (unfilled versus filled circles representing dysfunctional versus functional mtDNAs). (**B**) The mutualistic cooperation between mitochondria and eukaryotes is traditionally thought to be stabilized by PFF between eukaryotic host organisms and their mtDNAs/mitochondria. A host organism dominated by dysfunctional mtDNAs/mitochondria leaves fewer offspring (left) than a host organism dominated by functional mtDNAs/mitochondria (right). Due to mtDNA bottleneck, oocytes and offspring exhibit large variability in the level of dysfunctional mtDNAs/mitochondria they inherit from their mother. This facilitates PFF (see text). (**C**) Mitochondria dominated by functional mtDNAs replicate faster than those dominated by dysfunctional mtDNAs, even though dysfunctional mtDNAs replicate as fast as or faster than functional mtDNAs in the same mitochondrion. (**D**) PC between a focal cell and mitochondria/mtDNAs can occur through selective import, asymmetric segregation, or mitophagy. (**E**) Cells dominated by functional mtDNAs/mitochondria have a higher fitness than those dominated by dysfunctional mtDNAs/mitochondria. Grey crossbars: elimination; grey arrows: time.

mechanisms can operate in germ cells. For example, in birds and mammals, the majority of oocytes die via apoptosis, which has been hypothesized as a mechanism to purge dysfunctional mtDNAs/mitochondria (*Krakauer and Mira, 1999*). Furthermore, oocytes harboring dysfunctional mtDNAs/mitochondria are less likely to be fertilized (*Perez et al., 2000*). Thus, dysfunctional mtDNAs/mitochondria can be selected against because they lower the fitness of their host cells.

Finally, when we move one level further up, dysfunctional mtDNAs/mitochondria that have survived PC and PFFs at lower levels (*Figure 4C–E*) can be selected against *via PFF between host organisms and mtDNAs/mitochondria* (*Figure 4B*).

A model based on the Price equation predicts that mtDNA bottleneck (inheriting a small number of mtDNAs per mitochondrion or per germ cell) should facilitate PFF at the corresponding level. Indeed, mtDNA bottleneck has been observed during fly and mammalian oogenesis (*Koehler et al., 1991*; *Chinnery et al., 2000*; *Cree et al., 2008*; *Hill et al., 2014*) (Appendix note 27). This mtDNA bottleneck means that oocytes from a mother heteroplasmic for functional and dysfunctional mtDNAs will exhibit high variability in the frequency of functional mtDNAs that they inherit. Oocytes dominated by dysfunctional mtDNAs can be subsequently purged via apoptosis (Appendix note 28), thus creating a germline mtDNA filter that improves the prognosis of offspring.

## Conclusion

To reflect the hierarchical organization of biological systems, I introduce the concept of 'entity'. I have revised the definitions of PC and PFF so that they are orthogonal and so that they can operate on organismal or sub-organismal entities. PC requires that a focal cooperative entity can direct more benefits to cooperative over cheating partner entities despite their spatial equivalence. PFF suppresses cheaters that have escaped PC (if PC exists) via repeated interactions between entities in groups that, by chance, show high variability in cooperator frequency. By applying this revised definition, I show that sometimes, PFF at a lower level (e.g. between yucca flowers and moth offspring) may appear as PC at a higher level (between a yucca and moth offspring). PFF and PC can also act in conjunction at multiple levels to promote cooperation, as exemplified in the mitochondrion-eukaryote cooperation. This framework brings conceptual clarity for understanding the persistence of cooperation.

## Acknowledgement

I am grateful to the chance events that have induced this work. In 2013, Diethard Tautz kindly invited me to the EMBO EMBL Symposium 'New Model Systems for Linking Evolution and Ecology'. After my talk, Andrew Moore from BioEssays invited me to write about the evolution of cooperation. I decided to tackle the controversy raised by the very interesting Weyl et al. paper. I initially defined PC as being 'active' in the genetic sense (e.g. flower abortion is PC if a mutation can prevent such abortion). Reviewers were (rightly) not happy with this definition. The verdict was that I should write a review instead, a task that I was not too enthusiastic about given the many existing reviews. To better understand PC versus PFF, I subsequently went through many versions of ideas. During this time, Babak Momeni tirelessly served as my intellectual sounding-board. I thank my colleagues (Maxine Linial, Karen Craig, Chichun Chen, Robin Green, Björn Kafsack, Ben Kerr, Will Harcombe, and Suzannah Rutherford) and participants of the 2015 Microbial Population Biology Gordon Research Conference (Genya Frenkel, Wolfram Möbius, and Kevin Foster) for discussions and to Ben Kerr for *Figure 2*. Finally, I thank all reviewers (Jim Bull, Luke McNally, Diethard Tautz, Douglas Yu, and anonymous) for constructive criticisms. I dedicate this work to my *alma mater* Pomona College, Claremont, California, United States.

## Additional information

Competing interests

WS: Reviewing editor, *eLife.*

## Funding

| Funder | Grant reference number | Author |
|---|---|---|
| W.M. Keck Foundation | Distinguished Young Scholars Program | Wenying Shou |
| NIH Office of the Director | NIH Director's New Innovator Award DP2 OD006498-01 | Wenying Shou |
| Fred Hutchinson Cancer Research Center | New Development | Wenying Shou |

The funders had no role in study design, data collection and interpretation, or the decision to submit the work for publication.

## Author contributions

WS, Conception and design, Analysis and interpretation of data, Drafting or revising the article

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

# Appendix

### Note 1

Interactions between individuals are instances where an individual changes the physiology of another individual. Interactions can be mediated through, for example, diffusible chemicals (e.g. metabolites, antibiotics) or physical binding.

### Note 2

Excreting waste byproducts that happen to be useful to other species is not considered cooperation. Even though waste excretion may involve an energetic cost, not excreting waste is even more costly. Thus, waste excretion results in a net fitness benefit instead of a net fitness cost to the excretor.

### Note 3

Preys provide a costly benefit to predators. However, we do not consider preys to cooperate with predators when predators never reciprocate any benefit (e.g. helping preys to reproduce).

### Note 4

Kin cooperation can be homotypic (*Figure 1A*) or heterotypic (e.g. division of labor in an ant colony). Although mutualistic cooperation is generally heterotypic (*Figure 1B*), it can be homotypic (e.g. different species all decompose cellulose to usable glucose which is shared by all).

### Note 5

Although cheaters invest less than cooperators in producing costly benefits, cheaters are not necessarily more fit than cooperators. First, cooperators and cheaters can accumulate different mutations while adapting to their abiotic environment, making it possible for cooperators to sometimes outcompete cheaters (*Morgan et al., 2012*; *Waite and Shou, 2012*; *Asfahl et al., 2015*). Second, self-interest and the ability to cooperate can be genetically linked (*Foster et al., 2004*; *Dandekar et al., 2012*). For example in *Dictyostelium discoideum, dimA* couples the ability to cooperate (forming non-reproductive stalk) with self-interest (forming reproductive spores). Finally, the fitness difference between cheaters and cooperators can depend on the environment. For example, cooperative but not cheating yeast cells pay a cost to express cell-surface invertase which converts extracellular sucrose to usable monosaccharides. Rare cooperators are more fit than abundant cheaters. This is because a cooperator can privatize a very small amount of monosaccharides generated by its cell-surface invertase, and this benefit already exceeds the cost of cooperation when monosaccharides are scarce (*Gore et al., 2009*). On the other hand, rare cheaters are more fit than abundant cooperators. This is because abundant cooperators generate high levels of monosaccharides, and the privatization of monosaccharides by cooperators leads to only a negligible further increase in growth rate which can no longer offset the cost of invertase production (*Gore et al., 2009*). This frequency-dependent selection leads to the coexistence of cooperators and cheaters.

### Note 6

A group can have a well-defined boundary (e.g. cell membrane where the group comprises the cell and its internal endosymbionts), or not (e.g. microbes interacting with nearby neighbors in soil). Even though a group generally contains multiple individuals, a group can also contain a single individual.

### Note 7

Although Sachs et al. differentiated 'partner fidelity feedback' and 'partner choice' from 'kin fidelity' and 'kin choice', I have consolidated them to 'partner fidelity feedback' and 'partner choice' since partners can belong to different species, or genetically unrelated individuals of

the same species, or genetic relatives. One potential difference between homotypic kin cooperation and heterotypic mutualistic cooperation is that in kin cooperation, a focal individual can directly benefit from its own cooperative act (e.g. obtaining siderophores-$Fe^{3+}$ after releasing siderophores to scavenge environmental $Fe^{3+}$, *Figure 1A*). Such 'direct benefit' may be absent in mutualistic cooperation (e.g. plants making nectar solely for pollinators). However, direct benefit from cooperating with self may be considered as part of 'basal fitness' (the fitness of an individual when alone).

### Note 8

Too drastic a population bottleneck may destroy cooperation, because a critical density of cooperators is often required for effective reciprocation (*Shou et al., 2007*; *Koschwanez et al., 2011*).

### Note 9

Sachs et al. classified mechanisms such as 'sanctioning/policing' (punishing cheaters) and 'indirect reciprocity' (helping a partner who has helped others) under PC. The 'green-beard' mechanism, where an allele produces a perceptible trait which allows the recognition of and preferential treatment to those bearing the same trait, belongs to PC. For example, adhesive cells preferentially bind to other adhesive cells (*Figure 1C,iii*); cells harboring a toxin-antitoxin plasmid only kill cells without the plasmid (*Figure 3D*). Sachs et al. further noted that both PC and PFF can contribute to 'direct reciprocity' (helping a partner who has helped the individual itself) and 'kin selection' (helping genetic relatives even at a cost to self). Fletcher and Doebeli have also tried to unify all pro-cooperation mechanisms under 'positive assortment' where cooperators mostly receive cooperative acts from their interaction environment (*Fletcher and Doebeli, 2009*). However, this still leaves open the question what mechanisms can create positive assortment (*Fletcher and Doebeli, 2009*).

### Note 10

Another alternative criterion has been proposed to me: if and only if a mechanism (e.g. flower abortion) has evolved to exclude cheaters would it be considered PC. In reality, ascertaining why a mechanism has evolved is difficult unless the selective pressure is understood and the evolutionary history is available (e.g. experimental evolution). Thus, I do not favor this definition, since a definition should be 'physical as contrasted with psychological' and should strive to exclude 'psychological factors of preferences and decision making' (*Price, 1995*).

### Note 11

This definition requires that multiple entities exist. Thus, the entire earth ecosystem would not be considered an entity. However, this requirement should in general be easily satisfied.

### Note 12

Even though the birth of a tree depends on the birth of its parents, these birth events are temporally separable.

### Note 13

The contents of an entity can change. For example, in an entity comprising a yucca plant and its internal moth larvae, newly opened flowers receive new moth offspring as mature larvae exit.

### Note 14

Unlike concepts such as 'Darwinian individuals' (*Lewontin, 1970*), entities do not have to show heritable phenotypic variations that result in differential fitness. For example, flowers on the same yucca share the same genotype, and the death of a flower is not due to its genotype but to its pollinators'.

### Note 15

Cooperative and cheating partner entities may or may not be simultaneously present. For example, a focal cooperative entity can withhold interactions until it encounters a cooperative partner entity. PC operates as long as the focal individual would have favored cooperative over spatially-equivalent cheating partner entities should they be simultaneously present.

### Note 16

Consider the engineered yeast community (*Figure 3A*). Initially, the two partner populations [L− A+] and [L−] are spatially equivalent to the focal population [A− L+] due to their random spatial distributions. The preferential mixing of [A− L+] with [L− A+] instead of [L−] is driven by PFF not PC.

### Note 17

For example, PC may require a spatially-structured environment: Suppose that an imaginary [A− L+] cell could chemotax up an adenine gradient to interact with [L− A+] instead of [L−] despite their spatial equivalency. This PC would rely on adenine gradient whose formation requires a spatially-structured environment (*Figure 3C*). Additionally, the imaginary chemotactic [A− L+] cell would not be able to distinguish cooperative versus cheating partner cells in a clump.

### Note 18

To establish a mathematical model, one would quantify in monocultures of cooperators and cheaters their rates of birth, non-suicidal death, and suicidal death under favorable (low cell density) and unfavorable (high cell density) environments. One would also quantify the amount of toxin released by dying cooperator, and how toxin at various concentrations increase the death rate of cheaters. If in addition to reducing competition, cell death supplies metabolites to increase the survival of living cells, then these metabolites may need to be quantified as well. In a spatially-structured environment (PFF), one would also quantify the diffusion of toxins and metabolites. Such a model can be used to predict the dynamics of cooperator frequency.

### Note 19

To see how this works, imagine a yucca where flowers are not spatially separated. In this imaginary yucca with only one flower, depending on whether the one flower is retained or aborted, cooperative pollinators will be less fit than or as unfit as cheating pollinators, respectively.

### Note 20

Eukaryotic cells and their intracellular mitochondria interact in mutualistic manners (*McBride et al., 2006*). For example, mitochondria generate ATP via the TCA cycle and oxidative phosphorylation, carry out steps in the synthesis of several amino acids (*Ljungdahl and Daignan-Fornier, 2012*) and cofactors, and facilitate the maturation of essential cellular Fe-S proteins (*Lill and Kispal, 2000*). In multicellular organisms, mitochondria also serve as the site for apoptotic signaling and anti-viral signaling (*McBride et al., 2006*). Conversely, mitochondria rely on the host cell for survival, as evidenced by their highly reduced genomes and the ongoing transfer of DNA from mitochondria to the nucleus (*Ricchetti et al., 1999*; *Timmis et al., 2004*). On the flip side, mitochondria produce reactive oxygen species (ROS) which cause high mutation rate especially in mitochondrial genomes (mtDNAs), creating dysfunctional mtDNAs. ROS also cause cellular damage and organismal ageing (*Chomyn and Attardi, 2003*).

### Note 21

For example a cheating mtDNA in *S. cerevisiae* has replaced its genes with many copies of a DNA fragment containing the origin of replication, and is competitively superior to normal mtDNAs. This cheating mtDNA is defective in mitochondrial functions, and is deleterious to the host cell (*Blanc and Dujon, 1980*; *Taylor et al., 2002*; *Poole et al., 2012*). As another

example, in the aged human substantia nigra, the primary site of neurodegeneration in Parkinson disease, individual pigmented neurons often contain very high levels of clonally-expanded mutant mtDNAs containing various deletions (*Kraytsberg et al., 2006*; *Bender et al., 2006*). Because a neuron rarely undergoes mitosis, this age-dependent clonal expansion of a mutant mtDNA presumably occurs during the regular turnover of mitochondria and mtDNAs (*Menzies and Gold, 1971*) over the life time of a neuron, and likely reflects the fitness advantage of mutant over normal mtDNA.

### Note 22

For example in *S. cerevisiae* and female mammals, mutant and normal mtDNAs in a 'heteroplasmic' mixture can be transmitted to the next generation without any bias, depending on the mutation type (*Chinnery et al., 2000*; *Taylor et al., 2002*; *Freyer et al., 2012*).

### Note 23

Even though functional mtDNAs and functional mitochondria may not strictly correspond (e.g. mitochondria with functional mtDNAs may appear dysfunctional due to compromised vacuolar acidity [*Hughes and Gottschling, 2012*], and mitochondria with low levels of dysfunctional mtDNAs can still be functional), this correspondence largely holds.

### Note 24

For example in *S. cerevisiae*, cells with normal mtDNAs grow significantly faster than cells with dysfunctional mtDNAs, thus rendering dysfunctional mtDNAs/mitochondria less competitive (*Taylor et al., 2002*). In flies and mammals, an organism can even die at a young age if certain types of dysfunctional mtDNA reach sufficiently high levels (*Wallace, 1999*; *Hill et al., 2014*).

### Note 25

An eukaryotic cell generally contains a population of mitochondria, and each mitochondrion on average contains multiple mtDNAs (*Figure 4A*) (*Satoh and Kuroiwa, 1991*). Mitochondria undergo dynamic fusion and fission which mix organelle contents including mtDNAs (*Chan, 2006*). Thus inside a cell, a scenario similar to that modeled in the Price equation (*Figure 2*) occurs: mtDNAs are divided into spatially-separated mitochondria that interact through fusion (group dissolution) and fission (group formation). mtDNAs can cooperate with their host mitochondrion by, for example, generating the membrane potential required for resource import into mitochondrion, or cheat by not performing these tasks (*Clayton, 1992*). A model based on the Price equation predicts that frequent fusion will homogenize mtDNAs across mitochondria, and thus allow competitively-superior cheating mtDNAs to rise to high frequency even if fusion temporarily allows functional mitochondria to complement dysfunctional mitochondria. Too little fusion or fission is also predicted to disfavor cooperative mtDNAs, because in isolated groups, cheating mtDNAs will eventually arise from cooperative mtDNAs via mutations and outcompete cooperative mtDNAs.

### Note 26

For example, in mammalian tissue culture cells, compromised membrane potential of a mitochondrion causes PINK1 to be stabilized on mitochondrial membrane instead of being imported and rapidly degraded (*Jin et al., 2010*). Stabilized PINK1 degrades proteins required for mitochondrial fusion, and a blockage in mitochondrial fusion (*Lazarou et al., 2012*) attracts the mitophagy apparatus (*Narendra et al., 2008*; *Twig et al., 2008*).

### Note 27

For example, a mature mouse oocyte harbors mtDNAs on the order of $10^5$. The lack of new mtDNA synthesis and binary cell divisions cause a reduction in mtDNA to ~4000/cell before implantation (*Cree et al., 2008*). mtDNA copy number is further reduced to as low as 30, with a median of ~200, in primordial germ cells which eventually undergo meiosis and develop into oocytes or sperms (*Cree et al., 2008*).

**Note 28**

Strikingly, species where individuals give birth to fewer offspring tend to experience severer mtDNA bottleneck as well as apoptosis in a larger fraction of oocytes (*Krakauer and Mira, 1999*). This is consistent with the idea that if the number of offspring per individual is small, then selection for functional mtDNA in germ cells becomes critical (*Krakauer and Mira, 1999*).

