## [Decision Letter]

Thank you for submitting your work entitled "A revised criterion for differentiating mechanisms that promote cooperation" for peer review at *eLife*. Your submission has been favorably evaluated by Diethard Tautz (Senior Editor and Reviewing Editor) and two reviewers: James Bull and Luke McNally.

The reviewers have discussed the reviews with one another and the Reviewing editor has drafted this decision to help you prepare a revised submission.

Summary:

The paper addresses the ambiguity in distinguishing between two mechanisms for the evolution of cooperation: partner choice (PC) and partner fidelity feedback (PFF). The author suggests distinguishing cases of PC from PFF by specifying the requirement that PC allows cooperators to direct their cooperation to other cooperators instead of cheaters despite their "spatial equivalence".

We agree that the arguments made in this paper are topical and of broad interest, and think that with refinement this paper could help resolve ongoing debates that the author mentions and be an important contribution to the field. However, we have some issues with the paper as it currently stands, particularly with the clarity of presentation and how practically useful the criterion suggested would be.

The manuscript is very densely packed at present, jumping between multiple diverse examples and theoretical issues in individual paragraphs. We suggest that the author restructures the manuscript, moving many of the examples to tables or boxes so that they do not distract from the narrative of the paper. This would greatly increase the clarity of the paper. While keeping the main text short, you could make optimal use of the online presentation options through Lens.

Both referees address the clarity issue in similar, but also partly complementary ways. The relevant passages below are directly taken from the reports.

Essential revisions:

*Reviewer #1:*

As an example that presents just such a challenge, I'll offer a hypothetical case that Joel Sachs and I pondered some years ago. A plant allows nitrogen-fixing bacteria to colonize various (discrete) patches in its roots. Each colonized patch functions independently of others. If the bacteria provide sufficient nitrogen to the patch, the plant cells in the patch proliferate and automatically 'feed' the bacteria carbohydrates, and the bacteria then proliferate as well; the mix of bacteria and plant cells in the patch increase accordingly and become a functioning nodule that benefits the plant and the bacteria. However, if the bacteria do not provide nitrogen to the patch, the plant cells in that patch die and thus so do the bacteria that colonized the patch. Thus, at the level of the individual patch, the mechanism is partner fidelity feedback – cooperators proliferate because of the feedback from helping their partners. But at the level of the entire plant, the mechanism is (I hope) partner choice – because the plant has disproportionately rewarded cooperative bacteria. The attempt to discriminate mechanisms gets murky unless one acknowledges that different mechanisms are operating, depending on which 'entities' are being considered.

The paper acknowledges and offers such examples, and it suggests a resolution to the existing debate by modifying the existing 'definition' of partner choice. The formal definition appears to emphasize discrimination among spatially equivalent partners as the key. My sense is that the new definition in this paper does indeed 'work,' but it needs to be expanded/modified in a couple ways to be fully operational. First, there is only a definition of partner choice. Partner fidelity feedback should also be defined, even if to say it is everything except choice. Second, the definition is cryptic about its use of 'entity.' I think that the formal definition should be expanded to explain more about 'entities.' The text following the definition does so, but something needs to go in the definition, so that when the definition gets quoted in future work, 'entities' are formally explained. I emphasize this point about entities because I suspect much of the confusion and controversy in the field lies in the failure to appreciate the potential operation of different mechanisms at different levels in the individual (that multiple levels of entities may exist within one individual); thus one of the paper's main contributions may be to raise awareness about entities. Note that my hypothetical example above fits logically into the 'entity' paradigm – the root patches are one level of entity, the cells within the patches another level.

The foregoing comprises my main point – clear up the definition. However, I also think (strongly) that the paper is difficult to follow. It is of course challenging to write such a paper, because the topic is not intuitive, so a general audience needs to be given considerable background that is already well known within the field. Keeping the paper short limits the depth of this background before the paper dives into the new material. So such a paper will pose difficulties in the best of circumstances. But I nonetheless think the paper is not written to be widely accessible. Specifically, there is too much explanation embedded in details of biological systems to fully comprehend the (Abstract) points that constitute the substance of the paper. I think the paper would be more easily understood if it relied on hypothetical examples easily grasped by the reader: hypothetical examples can force the focus onto the key issues and avoid distracting details. The hypothetical examples can be borrowed from natural examples, but they should be structured to emphasize only the point being made. The description of natural examples could come late, once the main points are laid out with hypothetical examples.

Reviewer #2:

The author's example of cooperation between mitochondria and host cells illustrates how complicated the relationship between PFF and PC can be, particularly in hierarchically organised systems. However, this leads to the question of how useful is this distinction between PFF and PC for understanding the evolution of cooperation? Is this a meaningful debate? My general feeling is that the debate does matter and can help us identify mechanisms and genes underlying cooperation and mutualism. However, I think a general readership would greatly benefit from discussion of why the distinction between PFF and PC matters and isn't simply semantics. I would suggest adding this at the beginning of the paper to engage readers before the author outlines her revised criteria.

My final issue with the paper is that while the author has suggested a new criterion for distinguishing PC from PFF it is not immediately clear how this criterion could help design experiments to tell them apart. I would suggest that the author give at least one hypothetical example of how to design an experiment based on her suggested criteria in order to distinguish these two mechanisms. In particular, the author mentions the example of the conflicting views over whether the figure tree and figure wasp mutualism is maintained by PC or PFF. If the author could suggest a hypothetical experiment to distinguish between PC and PFF in this example it would greatly strengthen her case that this distinction is practically useful for understanding mechanisms of cooperation.

Both reviewers:

The supplementary material on the Price equation does not really contribute much. It would be useful if the Price equation could be written to separate partner choice from fidelity feedback. This paper by Kevin Foster uses a direct fitness model to do just this: http://zoo-kfoster.zoo.ox.ac.uk/sites/default/files/files/FosterWenseleersJEB2006.pdf. Reworking of this model into Price equation form could likely be used to address this point.

[Editors' note: further revisions were requested prior to acceptance, as described below.]

Thank you for resubmitting your work entitled "Redefine mechanisms that promote cooperation" for further consideration at *eLife*. Your revised article has been favorably evaluated by Diethard Tautz (Senior Editor and Reviewing Editor) and two reviewers. The manuscript has been improved but there are some remaining issues that need to be addressed before acceptance, as outlined below:

Reviewer 1 provides several comments on how to further improve the manuscript which you should address in your revised version. Reviewer 2 raises a further important point, namely the discussion of the question of intraspecific competition. The full comments of this reviewer are also attached. However, after discussion between the reviewers and the editor, we have come to the conclusion that including this into the current version of the manuscript would increase its length beyond its limits and would make it even more complex. Accordingly, we should like to ask you to drop this part entirely and reserve it for a possible follow-up paper in a more specialized journal. Hence, the comments of Reviewer 2 are only for your information.

Reviewer #1:

This is a revision; its predecessor was longer, far less focused and difficult to follow. The revision is now at a stage that is succinct and complete, but it still needs work. I think that the manuscript now needs to add structure and some clarification, as below. As before, I think the paper has a meaningful and necessary contribution to add to our understanding of the evolution of cooperation.

1) Make clear early what the paper is adding to the field. The paper should start with a clear statement of what is being done without going into detail about how it is being done. (The Abstract attempts to do so, but awkwardly.) Are PFF (partner fidelity feedback) and PC (partner choice) being redefined or just clarified? Is the result a 'reconciliation' or 'redefinition?' It seems that the big change is to recognize different organizational levels within an organism, such that PC can operate at one level and PFF operate at a different level; if so, then the paper recognizes PFF and PC as mechanisms that may operate within individuals as well as between them (which is new and useful).

2) The new definition of PFF is the complementary set to PC (i.e., PFF is everything that is not PC). This is not a good approach, because it runs the risk of including standard PFF with other mechanisms that are yet to be discovered. PFF should be defined as a true mechanism. Furthermore, if the new focus on entities is useful, it should lead to a clear definition of PFF.

3) The existing heading structure is good. I would add more. For each definition, I would have a separate heading: Definition: Entities; Definition: Partner Choice (etc.). Following the definitions of PC and PFF, I would have a heading 'How to demonstrate.' The paragraphs on how to demonstrate are useful and not something that would have occurred to me.

4) Something to contemplate: if PFF and PC work together within a system to maintain cooperation, do we want to recognize some additional basis for the evolution of cooperation above and beyond PFF and PC? I don't have an answer, but this might be something to think about.

*Reviewer #2:*

I think this revision of the manuscript has greatly improved its clarity, and all of my previous comments have been well addressed. However, I have one substantial revision that I feel is necessary before publication.

Reading the previous version of the manuscript I believed the author was primarily concerned with interspecies/heterotypic cooperation. However, from this revision and the author's responses I can now see that the ambition of this paper is wider, aiming to provide a framework for understanding both interspecies and intraspecies cooperation. I think this is fine, and the author generally makes a strong case. However, my major concern here is that while the relationship between the author's classification scheme and previous schemes for classifying interspecies cooperation is clear, there is insufficient discussion of the relationship between the author's classification scheme and previous classification schemes for the evolution of intraspecies cooperation. I hesitate to suggest new additions on a second round of review, but I think the author could address my points by adding a short subsection to the paper, and think that this is essential to support the lofty ambition of the paper.

The two most influential schemes for intraspecies cooperation are those provided by inclusive fitness theory and the "rule based" approach from Martin Nowak and other game theorists. In particular there have been two highly influential reviews classifying intraspecies cooperation from an inclusive fitness perspective that the author doesn't cite or discuss (West et al. 2007: http://www.zoo.ox.ac.uk/group/west/pdf/West_etal_07_CB.pdf & Lehmann & Keller 2006: http://onlinelibrary.wiley.com/doi/10.1111/j.1420-9101.2006.01119.x/abstract) and also one high-profile review from a game-theoretic perspective (http://ped.fas.harvard.edu/files/ped/files/science06_0.pdf?m=1425933699). I think the paper needs a short section comparing the utility of the authors scheme to these different perspectives to really convince the reader that this scheme makes progress relative to the classification schemes for intraspecies cooperation of inclusive fitness (classification based on direct vs indirect fitness effects) and game theoretic classifications (classification based on different games/rules). While I personally think that inclusive fitness theory is the most useful scheme, I think the author has a strong case that her scheme could be a strong competitor, but this needs to be made more explicit.

Related to this point, one thing that I think could help increase the readers intuition for the current scheme in the context of intraspecies cooperation would be adding an explanation (possibly as a table) of how previous classifications would fall within it. There is some of this throughout the manuscript, but it is not centralised in any way. I see the main strength of the author's scheme as being the separation of causes into two meaningful biological components, that of demography (partner fidelity feedback, which captures population structure, group augmentation, non-linear demographic feedbacks) and plasticity (partner choice, which captures direct, indirect and generalised reciprocity, punishment, policing, greenbeards, kin recognition). Stating this more explicitly would help relate the new scheme to previous schemes and highlight it's main strength – while more specific effects are difficult to partition (e.g. telling apart different forms of reciprocity), the author offers a simple way to separate the effects of demography and plasticity.

If the author can add a short section addressing these points I think it will help this paper have a wider-ranging impact on the field.

---

## [Author Response]

*[…] The manuscript is very densely packed at present, jumping between multiple diverse examples and theoretical issues in individual paragraphs. We suggest that the author restructures the manuscript, moving many of the examples to tables or boxes so that they do not distract from the narrative of the paper. This would greatly increase the clarity of the paper. While keeping the main text short, you could make optimal use of the online presentation options through Lens.*

I have chosen diverse examples to ensure that all examples can fit into this redefinition. Indeed, illustrating diverse examples and explaining conceptual nuances have made the writing challenging. In this revision, I started with abstract concepts, moved many examples to figures, and hid even more conceptual nuances into notes. I hope that I have succeeded in improving the flow.

*Both referees address the clarity issue in similar, but also partly complementary ways. The relevant passages below are directly taken from the reports. Essential revisions:*

Reviewer #1:

*As an example that presents just such a challenge, I'll offer a hypothetical case that Joel Sachs and I pondered some years ago. A plant allows nitrogen-fixing bacteria to colonize various (discrete) patches in its roots. Each colonized patch functions independently of others. If the bacteria provide sufficient nitrogen to the patch, the plant cells in the patch proliferate and automatically 'feed' the bacteria carbohydrates, and the bacteria then proliferate as well; the mix of bacteria and plant cells in the patch increase accordingly and become a functioning nodule that benefits the plant and the bacteria. However, if the bacteria do not provide nitrogen to the patch, the plant cells in that patch die and thus so do the bacteria that colonized the patch. Thus, at the level of the individual patch, the mechanism is partner fidelity feedback – cooperators proliferate because of the feedback from helping their partners. But at the level of the entire plant, the mechanism is (I hope) partner choice – because the plant has disproportionately rewarded cooperative bacteria. The attempt to discriminate mechanisms gets murky unless one acknowledges that different mechanisms are operating, depending on which 'entities' are being considered.*

You are exactly right. The same event (aborting non-functional nodules) can be interpreted as PC or PFF, depending on entities under consideration.

*The paper acknowledges and offers such examples, and it suggests a resolution to the existing debate by modifying the existing 'definition' of partner choice. The formal definition appears to emphasize discrimination among spatially equivalent partners as the key. My sense is that the new definition in this paper does indeed 'work,' but it needs to be expanded/modified in a couple ways to be fully operational. First, there is only a definition of partner choice. Partner fidelity feedback should also be defined, even if to say it is everything except choice. Second, the definition is cryptic about its use of 'entity.' I think that the formal definition should be expanded to explain more about 'entities.' The text following the definition does so, but something needs to go in the definition, so that when the definition gets quoted in future work, 'entities' are formally explained. I emphasize this point about entities because I suspect much of the confusion and controversy in the field lies in the failure to appreciate the potential operation of different mechanisms at different levels in the individual (that multiple levels of entities may exist within one individual); thus one of the paper's main contributions may be to raise awareness about entities. Note that my hypothetical example above fits logically into the 'entity' paradigm – the root patches are one level of entity, the cells within the patches another level.*

I am very grateful for these great points. I have expanded my definition and discussion on entity:

“A biological entity (entity) is a biological structure with a boundary such that the birth, or growth, or death of an entity is separable from that of other similar entities due to chemical or physical coupling within an entity and the lack of equivalent coupling between entities ^Note 10^. […] Thus, an entity (e.g. a figure tree-figure wasp entity) can contain smaller entities (e.g. the wasp larvae; the figure tree) which contain even smaller entities (e.g. figs) which contain even smaller entities (e.g. cells, mitochondria, mtDNA), reflecting the hierarchical organization of multicellular organisms ^Note 13^.”

I have added the following definition for partner fidelity feedback.

“I define PFF as the mechanism that promotes cooperation when PC does not exist or does not operate fully ^Note 16^ to exclude all cheaters. […] PFF can operate within a population of entities (e.g. homotypic cooperation in Figure 3) or between populations of entities (e.g. heterotypic cooperation in Figure 1).”

*The foregoing comprises my main point – clear up the definition. However, I also think (strongly) that the paper is difficult to follow. It is of course challenging to write such a paper, because the topic is not intuitive, so a general audience needs to be given considerable background that is already well known within the field. Keeping the paper short limits the depth of this background before the paper dives into the new material. So such a paper will pose difficulties in the best of circumstances. But I nonetheless think the paper is not written to be widely accessible. Specifically, there is too much explanation embedded in details of biological systems to fully comprehend the (Abstract) points that constitute the substance of the paper. I think the paper would be more easily understood if it relied on hypothetical examples easily grasped by the reader: hypothetical examples can force the focus onto the key issues and avoid distracting details. The hypothetical examples can be borrowed from natural examples, but they should be structured to emphasize only the point being made. The description of natural examples could come late, once the main points are laid out with hypothetical examples.*

Thanks for this suggestion. See my summary comments above.

Reviewer #2:

*The author's example of cooperation between mitochondria and host cells illustrates how complicated the relationship between PFF and PC can be, particularly in hierarchically organised systems. However, this leads to the question of how useful is this distinction between PFF and PC for understanding the evolution of cooperation? Is this a meaningful debate? My general feeling is that the debate does matter and can help us identify mechanisms and genes underlying cooperation and mutualism. However, I think a general readership would greatly benefit from discussion of why the distinction between PFF and PC matters and isn't simply semantics. I would suggest adding this at the beginning of the paper to engage readers before the author outlines her revised criteria.*

This is a good point. I have revised the text to stress the importance of defining orthogonal mechanisms. For example, in the first paragraph of the Introduction:

“How might cooperation evolve and persist despite the threat of cheaters? Identifying orthogonal mechanisms that independently promote cooperation will help us understand (and quantify) contributions from different sources. This will in turn move us closer toward predicting how persistent cooperation might be in a natural or artificial system.”

*My final issue with the paper is that while the author has suggested a new criterion for distinguishing PC from PFF it is not immediately clear how this criterion could help design experiments to tell them apart. I would suggest that the author give at least one hypothetical example of how to design an experiment based on her suggested criteria in order to distinguish these two mechanisms.*

I have added the following two paragraphs:

“This redefinition of PC leads to a conceptually simple experimental test. […] One can then test whether the focal entity preferentially directs benefit to the cooperative instead of cheating partner entity. If so, then PC operates.”

“To experimentally demonstrate the contribution of PFF in promoting cooperation, one will need to compare cooperator frequency when both PFF and PC are operative (e.g. in a spatially-structured environment) with when only PC is operative (e.g. in a well-mixed environment). […] In this case, one can compare cooperator frequency when PFF acts alone (in a spatially-structured environment where PC has been mutationally inactivated) with when neither PFF nor PC is operative (in a well-mixed environment with PC mutationally inactivated).”

In particular, the author mentions the example of the conflicting views over whether the figure tree and figure wasp mutualism is maintained by PC or PFF. If the author could suggest a hypothetical experiment to distinguish between PC and PFF in this example it would greatly strengthen her case that this distinction is practically useful for understanding mechanisms of cooperation.

In mutualisms between figure trees and figure wasps, PC between a figure tree and wasp offspring is identical to PFF between figs and wasp offspring. I have revised the text to hopefully make this clearer:

“We now return to the cooperation between figure trees and figure wasps. Selection against cheating pollinators can occur via PC between a focal figure tree and its internal pollinator offspring (Figure 4, left). […] Even though Sachs et al. and Weyl et al. respectively classify this cooperation as being stabilized by PC and PFF, my redefinition suggests that both are correct depending on entities under consideration.”

*Both reviewers:*

*The supplementary material on the Price equation does not really contribute much. It would be useful if the Price equation could be written to separate partner choice from fidelity feedback. This paper by Kevin Foster uses a direct fitness model to do just this: http://zoo-kfoster.zoo.ox.ac.uk/sites/default/files/files/FosterWenseleersJEB2006.pdf. Reworking of this model into Price equation form could likely be used to address this point.*

I have deleted the supplementary material on the Price equation. I did cite Kevin Foster’s work in my original version and this version. To potentially extend the Price equation, we note that ΔP¯, the change in community-wide cooperator frequency after one round of interactions (group formation, interaction, and reproduction), is determined by:

π¯ΔP¯=Cov(π,P)+Ave(πΔP)=β⋅Var(P)+Ave(πΔP). Here, a group with initial cooperator frequency of P will have final size π and final cooperator frequency of *P + ΔP* after one round of interactions, and average group size will be π¯. *Var(P)* is the variance of initial cooperator frequency *P* across all groups, and *β* is a correlation coefficient. We can simply equate the Price equation to contributions from PC and PFF:

π¯ΔP¯=Cov(π,P)+Ave(πΔP)=β⋅Var(P)+Ave(πΔP)=PFF+PC+π¯ΔP¯Exp, where ΔP¯Expis the expected change in community-wide cooperator frequency in the absence of PFF and PC. But this formulation seems a bit trivial.

If we look at each term of the Price equation, PC can contribute to *Var(P)* (e.g. yeast flocculation, Figure 1Ciii). PC can also contribute to Ave(πΔP) by diminishing the advantage of cheaters over cooperators in an interaction group (e.g. toxin-antitoxin, Figure 3). PFF relies on stochasticity during group formation to achieve *Var(P)*. The duration of groups will also affect both terms in the Price equation. If group duration is short, Cov(π,P)or β⋅Var(P) will be small due to limited amount of repeated interactions; if group duration is long, this term may also be small since groups that initially differed in cooperator frequency will eventually look alike as cheaters increase in frequency in all groups. Group duration will also affectAve(πΔP)by affecting final group size π as well as ΔP. Currently, I don’t see a biologically meaningful way of neatly partitioning the two terms in the Price equation into PC and PFF. This can be an interesting future project, but is beyond the scope of this article.

[Editors' note: further revisions were requested prior to acceptance, as described below.]

*Reviewer 1 provides several comments on how to further improve the manuscript which you should address in your revised version. Reviewer 2 raises a further important point, namely the discussion of the question of intraspecific competition. The full comments of this reviewer are also attached. However, after discussion between the reviewers and the editor, we have come to the conclusion that including this into the current version of the manuscript would increase its length beyond its limits and would make it even more complex. Accordingly, we should like to ask you to drop this part entirely and reserve it to a possible follow-up paper in a more specialized journal. Hence, the comments of Reviewer 2 are only for your information.*

Thank you for your consideration. I did address comments from Dr. McNally, since he raised good points about how to reach a wider readership, and since adding a few sentences did not take much extra space.

*Reviewer #1: 1) Make clear early what the paper is adding to the field. The paper should start with a clear statement of what is being done without going into detail about how it is being done. (The Abstract attempts to do so, but awkwardly.) Are PFF (partner fidelity feedback) and PC (partner choice) being redefined or just clarified? Is the result a 'reconciliation' or 'redefinition?' It seems that the big change is to recognize different organizational levels within an organism, such that PC can operate at one level and PFF operate at a different level; if so, then the paper recognizes PFF and PC as mechanisms that may operate within individuals as well as between them (which is new and useful).*

This is a great point. I have revised the title and the Abstract to shift the focus as per your suggestion. I have also changed “redefinition” to “revised definition” throughout the text.

Acknowledging selection at sub-organismal levels resolves controversy on pro-cooperation mechanisms

“Cooperators who pay a cost to produce publically-available benefits can be exploited by cheaters who do not contribute fairly. […] I also show that cooperation between multicellular eukaryotes and mitochondria is promoted not only by the traditionally-thought PFF between hosts and symbionts, but also by PFF and PC between sub-organismal entities.”

*2) The new definition of PFF is the complementary set to PC (i.e., PFF is everything that is not PC). This is not a good approach, because it runs the risk of including standard PFF with other mechanisms that are yet to be discovered. PFF should be defined as a true mechanism. Furthermore, if the new focus on entities is useful, it should lead to a clear definition of PFF.*

I have revised the definition of PFF as:

“Let us now consider potential mechanisms that promote cooperation when PC does not exist or does not operate fully ^Note 17^ to exclude all cheaters. One mechanism orthognal to PC is PFF. PFF requires the formation of interaction groups that vary significantly in cooperator frequency due to stochasticity and that last long enough so that cooperators benefit from their cooperative acts (Figure 3 right, Figure 3).”

*3) The existing heading structure is good. I would add more. For each definition, I would have a separate heading: Definition: Entities; Definition: Partner Choice (etc.). Following the definitions of PC and PFF, I would have a heading 'How to demonstrate.' The paragraphs on how to demonstrate are useful and not something that would have occurred to me.*

I have done so.

*4) Something to contemplate: if PFF and PC work together within a system to maintain cooperation, do we want to recognize some additional basis for the evolution of cooperation above and beyond PFF and PC? I don't have an answer, but this might be something to think about.*

I have added the following to the main text:

“Ideally, one would quantify processes involved in PC and PFF, and mathematically model how they contribute to cooperator survival. […] Otherwise, we will need to look deeper into potential causes of this mismatch.”

and the following to Figure 3 legend:

“To establish a mathematical model, one would quantify in monocultures of cooperators and cheaters their rates of birth, non-suicial death, and suicidal death under favorable (low cell density) and unfavorable (high cell density) environments. […] In a spatially-structured environment (PFF), one would also quantify the diffusion of toxins. Such a model can be used to predict the dynamics of cooperator frequency.”

*Reviewer #2: I think this revision of the manuscript has greatly improved its clarity, and all of my previous comments have been well addressed. However, I have one substantial revision that I feel is necessary before publication. Reading the previous version of the manuscript I believed the author was primarily concerned with interspecies/heterotypic cooperation. However, from this revision and the author's responses I can now see that the ambition of this paper is wider, aiming to provide a framework for understanding both interspecies and intraspecies cooperation. I think this is fine, and the author generally makes a strong case. However, my major concern here is that while the relationship between the author's classification scheme and previous schemes for classifying interspecies cooperation is clear, there is insufficient discussion of the relationship between the author's classification scheme and previous classification schemes for the evolution of intraspecies cooperation. I hesitate to suggest new additions on a second round of review, but I think the author could address my points by adding a short subsection to the paper, and think that this is essential to support the lofty ambition of the paper.*

This is a good point. I have added:

“Several conceptual frameworks have attempted to classify pro-cooperation mechanisms (Sachs et al., 2004; Nowak, 2006; Lehmann and Keller, 2006; West, Griffin and Gardner, 2007). […] For example in kin cooperation, Hamilton’s “kin discrimination” and “limited dispersal” (Hamilton, 1964) would correspond to PC and PFF, respectively.”

I also explain how tit-for-tat links to PFF:

“As an example of PFF, let’s consider “proportional tit for tat (pTFT)”, a game theory stradegy. […] pTFT at group size two becomes the standard TFT, and to survive cheaters, both requirements of PFF must be satisfied: repeated interactions in groups and large variation in the frequency of TFT across groups (Axelrod and Hamilton, 1981).”

*The two most influential schemes for intraspecies cooperation are those provided by inclusive fitness theory and the "rule based" approach from Martin Nowak and other game theorists. In particular there have been two highly influential reviews classifying intraspecies cooperation from an inclusive fitness perspective that the author doesn't cite or discuss (West et al. 2007: http://www.zoo.ox.ac.uk/group/west/pdf/West_etal_07_CB.pdf & Lehmann & Keller 2006: http://onlinelibrary.wiley.com/doi/10.1111/j.1420-9101.2006.01119.x/abstract) and also one high-profile review from a game-theoretic perspective (http://ped.fas.harvard.edu/files/ped/files/science06_0.pdf?m=1425933699). I think the paper needs a short section comparing the utility of the authors scheme to these different perspectives to really convince the reader that this scheme makes progress relative to the classification schemes for intraspecies cooperation of inclusive fitness (classification based on direct vs indirect fitness effects) and game theoretic classifications (classification based on different games/rules). While I personally think that inclusive fitness theory is the most useful scheme, I think the author has a strong case that her scheme could be a strong competitor, but this needs to be made more explicit.*

I found it quite amusing that West et al. criticized Nowak’s review in the following manner:

“First, we do not need to keep reinventing the wheel with more theoretical models that incorrectly claim to provide a new mechanism for the evolution of cooperation [12,97,98][…] This is illustrated by a recent review which

suggests five mechanisms for the evolution of cooperation [104]—three of these were just different ways of modelling the same thing (kin selection) [97,105–107], two were different forms of reciprocity.”

Yet, many mechanisms listed in West et al. Figure 2 may be regarded as kin discrimination (PC). I have now cited all these papers (see comment above). I have also added to note 7:

“One potential difference between homotypic kin cooperation and heterotypic mutualistic cooperation is that in kin cooperation, a focal individual can directly benefit from its own cooperative act (e.g. obtaining siderophores-Fe^3+^ after releasing siderophores to scavenge environmental Fe^3+^, Figure 1). […] However, direct benefit from cooperating with self may be considered as part of “basal fitness” (the fitness of an individual when alone).”

*Related to this point, one thing that I think could help increase the readers intuition for the current scheme in the context of intraspecies cooperation would be adding an explanation (possibly as a table) of how previous classifications would fall within it. There is some of this throughout the manuscript, but it is not centralised in any way. I see the main strength of the author's scheme as being the separation of causes into two meaningful biological components, that of demography (partner fidelity feedback, which captures population structure, group augmentation, non-linear demographic feedbacks) and plasticity (partner choice, which captures direct, indirect and generalised reciprocity, punishment, policing, greenbeards, kin recognition). Stating this more explicitly would help relate the new scheme to previous schemes and highlight it's main strength* –

*while more specific effects are difficult to partition (e.g. telling apart different forms of reciprocity), the author offers a simple way to separate the effects of demography and plasticity.*

I understand what you mean. However, if phenotypic plasticity is the ability of an organism to change its phenotype in response to changes in the environment, then plasticity is also seen in PFF: a focal cooperator will grow poorly near a cheating partner, and grow fast near a cooperative partner.

*If the author can add a short section addressing these points I think it will help this paper have a wider-ranging impact on the field.*

See comments to your first point.